# A Review on Endoscopic Ultrasound-Guided Radiofrequency Ablation (EUS-RFA) of Pancreatic Lesions

**DOI:** 10.3390/diagnostics13030536

**Published:** 2023-02-01

**Authors:** Fred G. Karaisz, Osama O. Elkelany, Benjamin Davies, Gerard Lozanski, Somashekar G. Krishna

**Affiliations:** 1Division of Gastroenterology, Hepatology, and Nutrition, Department of Internal Medicine, The Ohio State University Wexner Medical Center, Columbus, OH 43210, USA; 2Department of Internal Medicine, The Ohio State University Wexner Medical Center, Columbus, OH 43210, USA; 3College of Medicine, The Ohio State University, Columbus, OH 43210, USA; 4Department of Pathology, The Ohio State University Wexner Medical Center, Columbus OH 43210, USA

**Keywords:** pancreatic cystic lesion (PCL), pancreatic adenocarcinoma (PDAC), pancreatic neuroendocrine tumor (PanNET), endoscopic ultrasound-guided radiofrequency ablation (EUS-RFA), intraductal papillary mucinous neoplasm (IPMN)

## Abstract

The morbidity associated with pancreatectomies limits surgical options for high-risk patients with pancreatic neoplasms that warrant resection. Endoscopic ultrasound-guided radiofrequency ablation (EUS-RFA) offers a minimally invasive and potentially definitive means to treat pancreatic neuroendocrine tumors and precancerous pancreatic cystic lesions. In addition, EUS-RFA may play a role in the treatment and palliation of non-surgical cases of pancreatic adenocarcinoma. The efficacy of RFA appears to be further enhanced by systemic immunomodulatory effects. Here, we review current studies on the developing role of EUS-RFA in these pancreatic pathologies.

## 1. Introduction

Pancreatic adenocarcinoma continues to increase in incidence while it retains an abysmal 5-year survival rate ranging from 2% to 9% worldwide [1]. Hence, research in the early detection, characterization, and management of various pancreatic lesions has been a substantial focus within multiple disciplines. A host of potential therapies have been piloted for pancreatic lesions, including endoscopic ultrasound (EUS)-guided chemoablation [2], irreversible electroporation [3,4,5], and cryoablation [6].

Among the established techniques for eradicating solid tumors, such as esophageal adenocarcinoma [7,8,9,10] and hepatocellular carcinoma [11,12,13,14], includes radiofrequency ablation (RFA). RFA involves delivering a high-frequency alternating current that increases the temperature inside cells, which induces coagulative necrosis of the targeted tissues, leading to apoptosis [15]. RFA is also believed to have immunomodulatory effects [16]; it is postulated that the systemic immune response from the release of tumor-related antigens in RFA could contribute to a durable oncologic response [17,18,19]. In fact, small studies have shown improvement in survival among patients with unresectable pancreatic adenocarcinoma with palliative RFA [20,21].

In terms of its endoscopic applications, RFA has an established role in the treatment of Barrett’s esophagus with dysplasia and the prevention of esophageal adenocarcinoma [7,8,9,10]. However, its use with EUS in the context of pancreatic lesions is a newer but demonstrably safe concept [22,23,24]. RFA of pancreatic lesions using EUS guidance offers real-time imaging guidance and visualization that lends itself to immense precision and minimal invasiveness [22]. In this review, we will discuss EUS-RFA in pancreatic cystic lesions (PCLs), pancreatic adenocarcinoma, and neuroendocrine tumors (NETs) [25,26,27].

## 2. Devices Available for EUS-RFA

The two most studied EUS-RFA devices include the Habib EUS-RFA and the STARMed EUSRA RF. The Habib EUS-RFA is manufactured by EMcision Ltd., London, United Kingdom. It has a wire with a 1Fr active tip and generates a frequency of up to 480 kHz. Currently, the only United States Food and Drug Administration (FDA)-approved device for EUS-RFA is the STARMed EUSRA RF electrode from TaeWoong Medical USA. This single-use 19 gauge (G) RFA needle with an active tip length ranging from 5 mm to 15 mm connects to their VIVA RF generator (STARmed, Koyang, Korea). The EUS-RFA probe can deliver a current (400–500 kHz) that generates high temperatures electromagnetically, which can induce cell apoptosis and coagulative necrosis. This frequency is maintained continuously at a set wattage until complete ablation is achieved at the impedance value of 800 ohms. Once this impedance level is reached, the power output automatically cuts off, thus preventing further tissue damage. This system also includes the VIVA pump, which cools the electrode tip by circulating saline at a lower temperature to reduce tissue charring.

## 3. EUS-RFA in Pancreatic Adenocarcinoma

Pancreatic cancer is the third-leading cause of cancer mortality in the United States, with a 5-year survival rate of 11%. It is projected to account for the second-highest cancer-related deaths by 2030, behind lung cancer [28,29]. Nearly 90% of pancreatic malignancies are pancreatic ductal adenocarcinoma (PDAC), with a poor 1-year survival rate of 18% [30]. Patients with pancreatic cancer are classified into four groups based on the extent of the disease: resectable, borderline resectable, locally advanced, and metastatic. These classifications influence treatment options and predict outcomes.

For patients with locally advanced or metastatic disease, systemic therapy remains the treatment of choice. For patients with resectable or borderline resectable disease, surgical resection with neoadjuvant or adjuvant therapy is preferred. Currently, surgical resection represents the only potential curative means of treating pancreatic cancer [31]. Unfortunately, it is estimated that 80% to 85% of patients with pancreatic cancer have unresectable or metastatic disease [28,32]. Conceivably, EUS-RFA has multiple potential roles in the management of PDAC, including downsizing tumors to improve surgical candidacy, ablating unresectable tumors as part of a multi-modal treatment approach, or as a palliative measure [33].

There were initial concerns about the application of EUS-RFA in managing pancreatic lesions since the surrounding normal pancreatic tissue is particularly thermosensitive. Thermal ablation can lead to inflammation with fibrotic and cystic transformation. Among the potential adverse effects of pancreatic RFA include pancreatitis, gastric wall injury, bowel injury, pancreatic duct injury and stricture, bile duct injury and leak, adhesions, and peritonitis [34,35,36,37]. One of the largest and earliest prospective studies on ultrasound-guided (during laparotomy) RFA of locally advanced pancreatic adenocarcinoma was a surgical safety and feasibility study involving 50 patients by Girelli et al. Ultrasound-guided RFA during laparoscopic surgery was found to have a favorable safety profile, which was further improved by reduction of RFA temperature from 105 °C to 90 °C (correspondingly complications rates reduced from 24% to 4%, respectively) [38]. Partly due to the reduction in RFA temperature, early clinical studies of EUS-RFA found far fewer pancreatic complications with endoscopic RFA than with surgical RFA [22,23,24].

Table 1 summarizes published studies involving EUS-RFA for pancreatic cancer. Pilot studies by Song et al. on the feasibility and safety of EUS-RFA in unresectable pancreatic cancer demonstrated successful application in six patients. Two of these patients experienced abdominal pain relieved by analgesics, and the remainder did not have any further complications [39]. Arcidiacono et al. presented another feasibility and safety pilot study relevant to EUS-RFA; however, their group used a bipolar device known as a cryotherm probe (CTP), which combines RFA with cryogenic cooling. Overall, 16 of the 22 patients with unresectable tumors and undergoing neoadjuvant chemotherapy underwent radiofrequency heating and carbon dioxide cooling. In six patients, the CTP could not be passed through the gastric wall or tumor. Reported complications included minor duodenal bleeding in one patient and abdominal pain in three patients [6].

Further feasibility and safety studies of EUS-RFA in unresectable pancreatic cancer also reported secondary endpoints, such as radiological response. Scopelliti et al. performed a CT scan 30 days after completion of EUS-RFA in all 10 patients in their study, which found tumor size reduction in five of their patients and stable tumor size in the remaining five. Complication rates were again low as only 2 of the 10 patients had self-limited abdominal pain [40]. In similar studies, Crino et al. reported a mean tumor size reduction of 30% among 8 patients on CT scan 30 days post-procedure and no major adverse events with EUS-RFA [41]. In comparison, Wang et al. found a 20% reduction in tumor apparent diffusion coefficient on MRI after performing multiple-round EUS-RFA with low ablation power in 11 patients [42].

Long-term outcomes of EUS-RFA in the setting of unresectable pancreatic cancer were evaluated in more recent studies. Oh et al. published an observational prospective study on 22 patients, 14 with locally advanced disease and 8 with metastatic disease, who received EUS-RFA and subsequent gemcitabine-based chemotherapy. The median number of EUS-RFA treatments was five for these patients, as treatment had been repeated for radiologic tumor burden reduction. A complication rate of 3.74% was reported out of the total 107 procedures performed. Notably, they found a median overall survival of 24 months among patients treated with EUS-RFA [43]. In a prospective study on 10 patients (7 with locally advanced disease and 3 with metastatic disease), Thosani et al. reported a median overall survival of 20.5 months following the successful treatment of all the patients with EUS-RFA in conjunction with systemic chemotherapy. These patients received 1 to 4 EUS-RFA treatments based on “completeness of ablation”. Tumor progression was observed in 2 patients, while tumor regression was recorded in 7 patients. Two patients remained alive at 61- and 81-month follow-ups after initial diagnosis, the latter of which originally had metastatic PDAC but became a surgical candidate and underwent laparoscopic pancreaticoduodenectomy following extended chemotherapy and two EUS-RFA treatments [44].

**Table 1 diagnostics-13-00536-t001:** Selected publications: EUS-RFA for the treatment of pancreatic cancer.

Reference	Year	Diagnosis	Number of Patients (*n*)	Median Tumor Size (mm)	Results	Complications
Girelli et al. [38] **	2010	Locally advanced exocrine pancreatic cancer	50	35	CA19-9 decreased from 184 to 47 units/mL	24% abdominal; 2% mortality
Arcidiacono et al. [6] ***	2012	Locally advanced pancreatic cancer	22	35.7	Decrease in tumor size in 38%	18% abdominal pain, 6% duodenal bleeding
Song et al. [39]	2016	Stage 3 pancreatic cancer (4); stage 4 pancreatic cancer (2)	6	38	Successful application of EUS-RFA in all 6 patients	Mild abdominal pain (2 patients)
Scopelliti et al. [40]	2018	Unresectable non-metastatic PDAC	10	45	Decrease in tumor size (50%); stable disease (50%)	No complications (60%); ascites (20%); peripancreatic effusion (20%)
Crino et al. [41]	2018	PDAC (7); pancreatic head metastasis from renal clear cell carcinoma (1)	8	36	30% mean tumor ablation	Mild abdominal pain (3); no major AEs
Wang et al. [42]	2021	Locally advanced (7); metastatic (4)	11	28	Tumor size decrease (2); CA19-9 decrease (5); increased tumor ADC * value and 20% tumor ablation (1)	Mild abdominal pain (2); no deaths or major complications
Oh et al. [43]	2022	Locally advanced (14); metastatic (8)	22	38	95.5% treatment failure	Abdominal pain and peritonitis
Thosani et al. [44]	2022	Locally advanced (7); metastatic (3)	10	38	20.5 months median survival; tumor progression (2); tumor regression (6)	No major AE *; worsening abdominal pain in 55% of sessions

* AE: adverse events; ADC: apparent diffusion coefficient. ** Laparoscopic ultrasound-guided RFA. *** Use of a bipolar device that combines RFA with cryogenic cooling.

## 4. EUS-RFA in Pancreatic Cystic Lesions

Pancreatic cystic lesions (PCLs) encompass a range of lesions with varying malignant potential. Frequently found incidentally on cross-sectional imaging (CT or MRI), PCLs have a 5% to 35% prevalence that increases dramatically with age [45,46,47]. Distinguishing among the specific types of PCLs presents another important challenge, as prognosis varies widely; this has called for novel methods for the differentiation of PCLs, such as EUS through-the-needle biopsy [48] and endomicroscopy [49]. Broadly, PCLs are divided into two groups: mucinous cysts and non-mucinous cysts. Mucinous cysts, which account for approximately 61% of PCLs [50], carry an elevated risk for malignant transformation. They include intraductal papillary mucinous neoplasms (IPMNs) and mucinous cystic neoplasms (MCNs). IPMNs constitute the majority of mucinous cysts and have the greatest risk for malignant transformation [51,52]. IPMNs are classified based on their location (Figure 1): main duct (MD), branch duct (BD), or mixed duct (a combination of two former categories) [53]. A host of other characteristics, including cytology, morphology, and biochemical features, are used to further stratify these IPMNs. BD-IPMN is the most common type, and the risk of malignant transformation with increasing size (greater than 3 cm) ranges from 12% to 25% [54]. BD-IPMN is amenable to EUS-RFA treatment (Figure 1 and Figure 2).

Based on multiple criteria, pancreatic cystic lesions with “high-risk stigmata” of malignancy and some with “worrisome features” are considered for treatment. The current standard of treatment for these precancerous PCLs is surgical resection [46,55]. Almost two-thirds of these surgeries require pancreaticoduodenectomy, as most of these neoplasms are found in the head of the pancreas [56,57]. While mortality for this highly-invasive surgery has improved over the last 20 years, the morbidity and complications have remained frustratingly constant [58,59]. Even minimally invasive surgical means of laparoscopic robotic-assisted pancreaticoduodenectomies have not been clearly shown to reduce operative morbidity [60,61].

Emerging modalities in the non-surgical management of PCLs include endoscopic techniques, the most studied of which are EUS-guided alcohol ablation and EUS-chemoablation [25,26]. EUS-guided alcohol ablation is associated with higher rates of pancreatitis, presumably from alcohol extravasation in the surrounding pancreatic parenchyma, compared to chemoablation with paclitaxel [62,63]. EUS-RFA is a newer option that takes advantage of the surgical efficacy of RFA in solid tumors. Of note, while all endoscopic strategies show potential, they have not yet been proven to reduce the risk of high-grade dysplasia or pancreatic cancer [46,64]. Hence, multicenter clinical trials are needed to validate the clinical benefit of these methods.

Among the first pilot studies of EUS-RFA for pancreatic cystic lesions, Pai et al. reported a multicenter safety and feasibility study on eight patients (six with PCLs and two with PanNETs) that showed complete resolution or ~50% reduction in the size of pancreatic cystic lesions for all of the patients with PCLs without any serious adverse events [65]. Of the patients with PCLs, four had MCNs, one had an IPMN, and one had a microcystic adenoma. Two patients had mild abdominal pain that resolved within three days. A recent single-center study by Younis et al. corroborated these findings, reporting complete radiologic response of PCLs to EUS-RFA in three of five patients (60%) and inadequate response in the remaining two patients [66]. In this study, 12 patients in total were treated with EUS-RFA (four had IPMNs, one had a MCN, and the remainder had PanNETs).

Long-term outcomes of EUS-RFA for precancerous PCLs have not been established. However, a prospective multicenter trial by Barthet et al. that originally focused on safety involving 14 patients with PanNETs and 17 patients with PCLs provides some insight. Subjects were followed for 12 months [67] and then a total of three years in a subsequent study [68]. The 17 patients with PCLs included 16 IPMNs and 1 mucinous cystadenoma (mean diameter = 29.1 mm; 10 head, 4 body, and 3 tail of the pancreas). In this study, the patients first underwent EUS-guided fine needle aspiration of the PCLs for cyst aspiration and then EUS-RFA with an 18-gauge RFA needle (STARmed, Koyang, Korea), applying 50W until an impedance of 100 Ohms was achieved. At 12 months, of the 17 patients with PCLs, 11 had complete resolution, and 1 had a decrease in diameter > 50%.

In the long-term (3-year) follow-up study, among 15 of the 17 patients with PCLs (two died due to unrelated causes), Barthet et al. found that 6 had continued complete resolution, and 4 had a decrease in diameter > 50%, representing an overall significant response of 67%. Three patients had a recurrence of a small cyst ranging from 4 mm to 6 mm [68]. This study suggests that overall long-term outcomes of EUS-RFA for precancerous PCLs are promising and further results are detailed in Table 2. The authors also theorized that the systemic immune response due to RFA could have been partially responsible for the long-term response.

Of note, this group reported a 10% complication rate in the overall study. Major complications included one patient with acute pancreatitis, one patient with jejunal perforation, and one patient with pancreatic duct stenosis. The authors found that the complication rate appeared to have been mitigated with rectal diclofenac, procedural intravenous antibiotics, and removal of the fluid content of the PCL by aspiration prior to RFA (complication rate decreased to 3.5% after the changes with no further occurrences of major complications) [67].

## 5. EUS-RFA in Neuroendocrine Tumors

Pancreatic neuroendocrine tumors (PanNET) arise from the endocrine tissues of the pancreas and comprise less than 5% of all pancreatic cancers [69,70]. PanNETs tend to be less aggressive than PDAC, yet they can be metastatic to the liver at a diameter ≥ 2 cm [71]. There are two groups of PanNETs: functional and nonfunctional. Functional PanNETs are characterized by the over-secretion of various hormones and can be associated with a range of clinical syndromes. Functional PanNETs include insulinomas, gastrinomas, glucagonomas, and somatostatinomas. Nonfunctional PanNETs comprise the majority of PanNETs (up to 75.4%) [72] and share histologic and pathologic properties with functional PanNETs but do not oversecrete hormones. It is important to note that the goal of treatment for functional PanNETs is to eliminate the neuroendocrine tumor cells in order to cease the associated hormonal hypersecretion, while the considerations for the treatment of nonfunctional PanNETs are more complicated because it is rooted in predicting and preventing further growth and advancement of the tumor [73].

Similar to precancerous PCLs, PanNETs (functional or ≥2 cm in size) are often treated by surgical resection via pancreaticoduodenectomy or distal pancreatectomy [74,75]. However, given their variability in clinical presentation and tumor characteristics, the optimal surgical strategy for PanNETs is not completely clear. In regards to PanNETs, the risk of a highly invasive surgery for a tumor that is not often aggressive can be difficult to justify in certain populations [76]. Some guidelines recommend that patients with PanNETs who are unfit for surgery should receive annual imaging for lesions with a size less than 10 mm and biannual imaging for 10 mm to 20 mm [52]. These considerations place the potential endoscopic treatments of tumor ablation for PanNETs in a unique light. In fact, a recent retrospective study comparing the safety and long-term outcomes of EUS-ethanol ablation and surgery among 188 patients showed comparable 10-year overall and disease-specific survival rates, while the EUS branch had fewer complications and shorter hospital stays [77].

EUS-RFA has also been described in multiple studies for the ablation of PanNETs (Table 3) and is poised to become the standard of care among functional PanNETs [73]. Among the earliest feasibility and safety studies for EUS-RFA of PanNETs, Rossi et al. [27] carried out a prospective study on 10 patients who were diagnosed with PanNETs (3 of them with functional PanNETs). The mean tumor diameter was 1.6 cm; 7 were located in the head of the pancreas and 3 in the tail. Ultrasound-guided RFA was conducted percutaneously in 7 patients, endoscopically in 1 patient, and intraoperatively in 1 patient. Successful ablation was achieved in all 10 patients after the RFA procedures. During a median follow-up of 34 months, no recurrences were observed. Complications include 3 incidences of acute pancreatitis, 2 of which developed fluid collections requiring ultrasound-guided drainage and endoscopy.

Similar successful small studies and case reports and series were reported with promising findings in terms of efficacy, feasibility, and safety (Table 3) [66,78,79,80,81,82,83]. Choi et al. [81] described a case series on seven patients with nonfunctional PanNETs (mean size of 20 mm, one 12 mm insulinoma). In the nonfunctional PanNET group, one case of abdominal pain and one case of pancreatitis occurred; no other adverse events were observed. Five out of the seven patients (71.4%) had a complete response. Thosani et al. [82] described EUS-RFA being used to successfully ablate functional PanNETs in three patients (two insulinomas and one VIPoma). Symptoms resolved in all patients after the procedure. A more recent retrospective study at 2 tertiary centers performing EUS-RFA in pancreatic insulinomas also showed rapid symptom improvement among 7 patients, 6 (85.7%) of which showed complete response on cross-sectional imaging and remained asymptomatic at 21-month follow-up [84].

More recently, larger multicenter studies on the EUS-RFA of PanNETs substantiated these earlier studies regarding efficacy, feasibility, and safety. Barthet et al. reported a prospective study of 28 patients undergoing EUS-RFA of either PanNETs or PCLs. In the PanNET arm, 12 patients had a total of 14 nonfunctional PanNETs (mean size = 13.4 mm). The EUS-RFA of these 14 PanNETs resulted in 12 with a complete response at 12-month follow-up (represents a significant response (>50% decrease in size) with a rate of 85.7%) [67]. This significant response rate of 85.7% was consistent at a 3-year follow-up [68]. Oleinikov et al. [85] reported on 18 patients from two tertiary centers, 7 of which had insulinomas while the remainder had nonfunctional PanNETs. There were 27 lesions in total, with a mean size was 14.3 mm. Complete response was observed in 26 of the 27 lesions (96%), and no complications were observed during the procedure. Additionally, no recurrences were reported after a mean follow-up of 8.7 months.

**Table 3 diagnostics-13-00536-t003:** Characteristics and Findings of Studies of EUS-RFA for pancreatic neuroendocrine tumors.

Reference	Year	# of Patients (*n*)	Size of Tumor (mm)	Complete Response (%)	Complications (%)	Recurrences (%)
Rossi et al. [27]	2014	10	16	100%	30% (3 incidences of pancreatitis)	0%
Armellini et al. [86]	2015	1	20	100%	0%	0%
Pai et al. [65]	2015	2	27.5	100%	0%	0%
Lakhtakia et al. [78]	2016	3	17	100%	0%	0%
Waung et al. [79]	2016	1	18	100%	0%	0%
Bas-Cutrina et al. [80]	2017	1	10	100%	0%	Not specified
Choi et al. [81]	2018	8	19	75%	25% (1 case of abdominal pain and 1 case of pancreatitis)	Not specified
Thosani et al. [82]	2018	3	Not specified	100%	Not specified	Not specified
de Mussy et al. [83]	2018	1	18	100%	0%	0%
Barthet et al. [67]	2019	12	13.1	85%	14% (1 case of pancreatitis, 1 case of pancreatic duct stenosis)	Not specified
Oleinikov et al. [85]	2019	18	14.3	96%	0%	0%

## 6. Post-RFA Durable Response

Multiple publications have suggested mechanisms for a systemic immune response elicited by RFA, which could explain findings of delayed response of treated PCLs [68] and even improvement in survival among patients with unresectable pancreatic adenocarcinoma with palliative RFA [20,21,43,44,87]. It is postulated that the immunologic effects of the hyperthermia induced by RFA can cause an “abscopal effect” [88]. Similar observations in the RFA of other solid tumors, including hepatocellular carcinoma [89,90,91], are substantiated by immunologic studies.

The improved survival benefit from RFA in hepatocellular carcinoma can possibly be attributed to local and systemic immune effects [92]. Specifically, the release of highly immunogenic intracellular components from hyperthermic tumor destruction by RFA, including heat-shock proteins [93,94], activates local tumor immune cells, such as myeloid dendritic cells [95,96]. Heat shock proteins (HSPs) are chaperone proteins involved in antigen presentation to the major histocompatibility complex 1 (MHC-1) of dendritic cells. Post-RFA treatment, elevated levels of HSP-70 expression were found in residual cancer cells, leading to their growth via the AKT-mammalian target of rapamycin (mTOR) signaling pathway. The combination of RFA with an mTOR inhibitor has been shown to reduce tumor growth [97].

In mouse cancer models, RFA resulted in the increased recruitment of cytotoxic T-cells [98] among treated mice with colon cancer and melanoma and helper T-cells [18] among treated mice with urothelial carcinoma, suggestive of an adaptive immune response. Compared to untreated mice, there was a significant elevation in dendritic cell infiltration in the tumor microenvironment of treated mice. Moreover, mice treated with RFA or intratumoral dendritic cell treatment had better control of tumor progression compared to the untreated group [18]. In fact, Dromi et al. observed an abscopal effect by RFA, as the complete elimination of the established tumor was observed despite only partial RFA ablation of the tumor. This group speculated that the development of a significant zone of sublethal heating within the tumor margin after RFA was crucial for this response, as inflammatory factors were measured to be particularly elevated within this zone [18].

Similar immunological studies are being undertaken for the RFA of pancreatic lesions [88], some of which have suggested an adaptive immune response due to the effects of RFA. Giardino et al. reported a significant increase in cytotoxic T-cells and dendritic cells between days 3 and 30 among 10 patients following RFA for locally advanced pancreatic cancer. In addition, interleukin-6 (IL-6), a proinflammatory cytokine, was markedly elevated among these patients on day 3. By day 30, dendritic cell levels continued to be elevated, while IL-6 levels returned to baseline [16]. Taken together, RFA appears to elicit more than a simple inflammatory response but may have sustained effects on the adaptive immune system that could confer a “durable response” upon treatment by EUS-RFA. While most of these studies center around malignant tumors, it is conceivable that these immune effects could be extrapolated for PanNETs and precancerous PCLs.

## 7. Discussion

There are numerous challenges in diagnosing and managing pancreatic pathologies, as highlighted most dramatically by the continued poor outcomes in pancreatic adenocarcinoma. Endoscopic ultrasound-guided techniques, such as EUS-RFA, have been shown to be a safe technique with the potential to have established roles in the treatment of pancreatic lesions [22]. As definitive management of PCLs, PanNETs, and PDAC all involve highly invasive surgery, the minimally invasive EUS-RFA has the potential to meet a previously unfilled demand.

RFA has been successfully utilized in dysplasia and solid tumors in multiple facets of gastroenterology and oncology, including esophageal adenocarcinoma [7,8,9,10] and hepatocellular carcinoma [11,12,13,14]. Endoscopic ultrasound enables the application of RFA in the pancreas, which offers immense precision and minimal invasiveness. In addition, it is believed that RFA elicits a systemic immunological response, which could contribute to a durable response [16,17,18,19].

The safety profile of EUS-RFA has been explored in several animal and human studies, which have yielded promising results (Table 1 and Table 3). The most common complications reported were abdominal pain and acute pancreatitis [22]. The safety and efficacy of RFA in the pancreas were first studied in surgical procedures; while overall complication rates were low, serious complications had been reported [99,100]. Complication rates due to RFA were further mitigated with the reduction of the RFA electrode tip temperature, as demonstrated by Girelli et al., who observed a decrease from 24% to 4% in complications upon reducing the RFA temperature from 105 °C to 90 °C [38]. Another significant change when comparing early surgical and EUS-RFA studies for safety is the development of the internal cooling system found in EUS-RFA probes [33]. The internal cooling system has been shown to limit ablation depth at varying power settings in porcine models [101].

The efficacy of EUS-RFA for various pancreatic lesions still needs to be elucidated, given the small sample size and lack of long-term follow-up in the currently reported studies. Perhaps the most promising data are within the use of EUS-RFA for PanNETs. Several studies suggest that EUS-RFA can achieve a complete response in 75% to 100% of patients (Table 3). A recent review of 12 studies summarized that the overall effectiveness rate of EUS-RFA on PanNETs was 96% among 61 patients [102]. Notably, this was consistent in a similar systematic review for the EUS-RFA of insulinomas, which involved 35 case reports and case series totaling 75 patients [103]. Some limitations to these studies in EUS-RFA for PanNETs include nonstandardized techniques as well as a lack of an agreed-upon definition of a complete response. Randomized control trials with longer periods of follow-up would be necessary to build upon these studies, which unfortunately can be limited due to the rarity of PanNETs.

EUS-RFA in pancreatic cancer has multiple potential roles. It may be the only definitive option for early pancreatic cancer in patients who are not amenable to surgery, or it can be paired with systemic chemotherapy, as RFA may improve the efficacy of neoadjuvant treatment with increased conversion to resectable disease [21]. EUS-RFA may also be a palliative treatment in unresectable pancreatic cancer. Some small studies have demonstrated improved survival when RFA was paired with current palliative measures [20,21,44]. Similarly, for EUS-RFA of PanNETs, randomized control trials studying survivability and long-term outcomes are necessary to further characterize these early findings.

EUS-RFA in precancerous PCLs is a particularly attractive option given the challenges in diagnosing with certainty the type of pancreatic neoplasms that need resection and the morbidity associated with pancreaticoduodenectomies. EUS-RFA appears to be safe for this indication [65,67], and there is some evidence that PCLs may respond to RFA long-term [68]. On 3-year follow-up for PCLs treated with EUS-RFA, the significant response after the initial treatment was found to be 67% (Table 2). It is conceivable that repeat treatment could enhance these outcomes further. Again, randomized control trials are necessary to establish the role of EUS-RFA in precancerous PCLs. Additional studies could also be carried out to compare the efficacy of chemoablation to RFA for precancerous PCLs.

In summary, EUS-RFA is an emerging modality for various pancreatic lesions that can spare patients from highly invasive surgeries. EUS-RFA will likely have defined roles in the treatment of precancerous PCLs, PanNETs, and PDAC in the next decade.

## Figures and Tables

**Figure 1 diagnostics-13-00536-f001:**
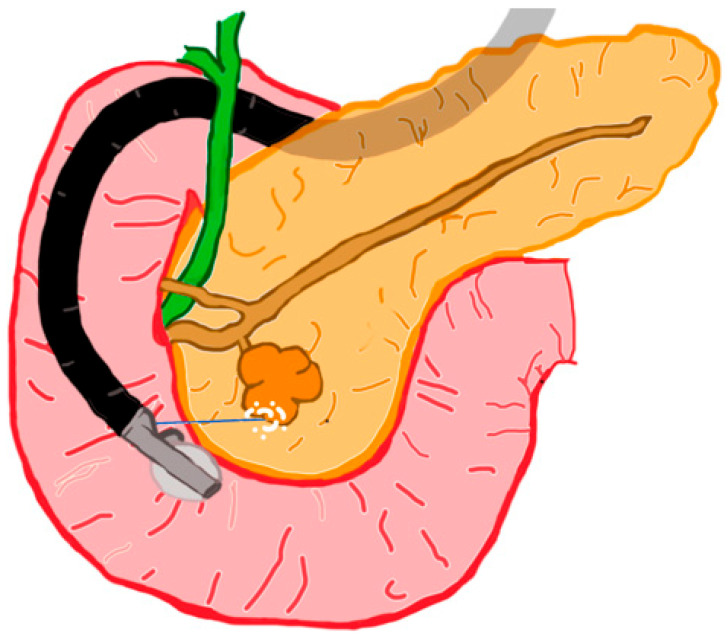
Schematic of EUS-RFA of a BD-IPMN.

**Figure 2 diagnostics-13-00536-f002:**
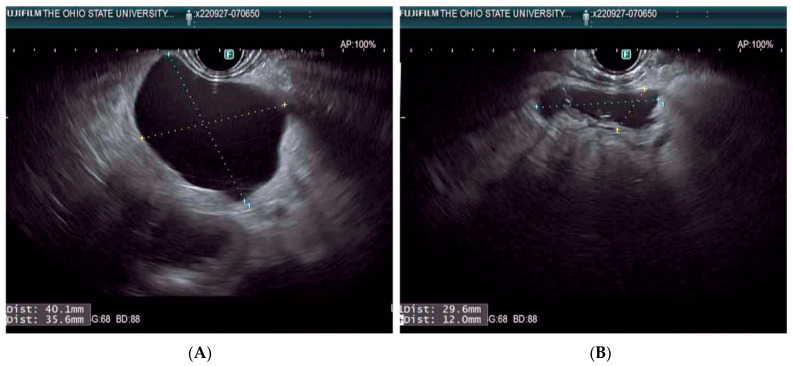
EUS-RFA of a BD-IPMN. Panel (**A**): pre-ablation, Panel (**B**): post-ablation.

**Table 2 diagnostics-13-00536-t002:** Summary of response to EUS-RFA in PCLs (Barthet et al., 2021) [68].

	Follow-Up Duration	
PCL, *n* = 17	12 months (*n* = 17)	3 year (*n* = 15)
Complete resolution	11 (~65%)	6 (~40%)
Significant response = decrease in size > 50% (includes complete resolutions)	12 (~71%)	4 (~27%)

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
