# Peer review of "A Review on Endoscopic Ultrasound-Guided Radiofrequency Ablation (EUS-RFA) of Pancreatic Lesions"

_diagnostics, 2023, doi:10.3390/diagnostics13030536_

Round 1

Reviewer 1 Report

The current manuscript reviews endoscopic ultrasound-guided radiofrequency ablation (EUS-RFA) for pancreatic neoplastic lesions. This is an interesting topic, although solid evidence from clinical trials is sparse. The contemporary literature is cited and discussed. The manuscript is well written and the discussion is balanced. It would be helpful to include a short paragraph about alternative local ablative therapies such as IRE, stereotactic radiation, and others.

Author Response

The current manuscript reviews endoscopic ultrasound-guided radiofrequency ablation (EUS-RFA) for pancreatic neoplastic lesions. This is an interesting topic, although solid evidence from clinical trials is sparse. The contemporary literature is cited and discussed. The manuscript is well written and the discussion is balanced. It would be helpful to include a short paragraph about alternative local ablative therapies such as IRE, stereotactic radiation, and others. 
  • This was also mentioned by another reviewer. Upon our review of the literature, studies comparable to the prospective trials on EUS-RFA for IRE and EUS-chemoablation are now mentioned in the introduction. Thank you so much for your suggestion.

Reviewer 2 Report

This is a comprehensive review on the role of EUS-RFA for the treatment of pancreatic lesions. The topic is very interesting and rapidly evolving. The paper is clear and well-written. 

I have a few suggestions:

- Please mention all the devices used so far for EUS-RFA, not only the one currently approved by FDA.

- Please, change PNET to PanNET

- For PanNETs, describe separately the results for functional and non functional tumors. The objective is different in case of functional or nonfunctional tumors, as well the definition of clinical efficacy. 

- Two recent papers on the treatment of PanNETs are missing: PMID 24963025 and PMID: 34902374 

- Other techniques for local ablation of pancreatic tumors are available. Please mention such techniques and cite PMID: 30505967

- Indications for the treatment of non-functional PanNETs have been comprehensively summarized in PMID: 31249164. Please mention and cite this study.

- A recent study compared EUS-ethanol injection and surgery for the treatment of nonfunctional PanNETs (PMID: 36400239). Despite this study used EUS-ethanol injection, I think it deserve to be mentioned.

- The importance of obtaining a specific diagnosis before the ablation of pancreatic cysts should be underlined. Doing so, cite PMID: 35451041

Author Response

- Please mention all the devices used so far for EUS-RFA, not only the one currently approved by FDA. 

  • We added a few lines on the Habib EUS-RFA, in addition to the STARMed EUSRA RF, which are the most tested based on the studies we are reviewing.

- Please, change PNET to PanNET. 

  • This change was made for our paper in its entirety.

- Other techniques for local ablation of pancreatic tumors are available. Please mention such techniques and cite PMID: 30505967. 

  • Another reviewer made a similar comment and we have incorporated both this paper and another into our introduction. Thank you for this citation!

- Indications for the treatment of non-functional PanNETs have been comprehensively summarized in PMID: 31249164. Please mention and cite this study.

  • We now include this citation, which is an excellent summary of perspectives of EUS-RFA on functional vs nonfunctional PanNETs, in our neuroendocrine section. 

- For PanNETs, describe separately the results for functional and non functional tumors. The objective is different in case of functional or nonfunctional tumors, as well the definition of clinical efficacy. 

  • We agree with this point completely and have used one of the papers you had suggested as a resource to mention this fact. 

- Two recent papers on the treatment of PanNETs are missing: PMID 24963025 and PMID: 34902374 

  • We could not locate and review the first paper mentioned here. The second paper now added to our review, but we did not include it in our table since it is a retrospective analysis. 

- A recent study compared EUS-ethanol injection and surgery for the treatment of nonfunctional PanNETs (PMID: 36400239). Despite this study used EUS-ethanol injection, I think it deserve to be mentioned.

  • We agree that this is a substantial retrospective study that is now mentioned in our introduction to EUS-RFA of neuroendocrine tumors section. 

- The importance of obtaining a specific diagnosis before the ablation of pancreatic cysts should be underlined. Doing so, cite PMID: 35451041

  • We included a line and publications on the methods of diagnosis, including the paper you referred us and another by our PI regarding endomicroscopy. Thank you for this suggestion!